# Defects Detection and Identification in Adhesively Bonded Joints between CFRP Laminate and Reinforced Concrete Beam Using Acousto-Ultrasonic Technique

Cheikh A. T. Sarr [1,*], Sylvain Chataigner [1], Laurent Gaillet [1] and Nathalie Godin [2]

1 Structures Métalliques et à Câbles (SMC), Dept Matériaux et Structures (MAST), Université Gustave Eiffel, Route de Bouaye, 44341 Bouguenais, France
2 MATEIS, UMR5510, INSA Lyon, Université Claude Bernard Lyon 1, 69621 Villeurbanne, France
* Correspondence: cheikh.sarr@univ-eiffel.fr

**Abstract:** Adhesively bonded composite reinforcements have been increasingly used in civil engineering since the 1980s. They depend on the effective transfer of forces throughout the adhesive joint that may be affected by defects or damages. It is therefore necessary to provide methods to detect and/or identify these defects present in the bonded joints without affecting their future use. This should be carried out through nondestructive methods (NDT) and should be able to discriminate the different types of defects that may be encountered. The acousto-ultrasonic technique shows good potential to answer to this challenge, as illustrated in recent studies led on small-scale model samples. In this paper, we assess the robustness of this methodology on larger scale samples using reinforced concrete beams (RC beam), that is a mandatory step prior to on-site applications. A mono-parametric analysis allows the detection of all types of defects using a simple criterion set. For the identification, it was necessary to conduct a data-driven strategy by means of a Principal Component Analysis (PCA) and a random forest (RF) method used from extracted parameters.

**Keywords:** acousto-ultrasonic; non-destructive technique (NDT); adhesively bonded joint; diagnostic; data-driven model; PCA; random forest

## 1. Introduction

Thanks to their ultra-light weight and durability, composite materials are increasingly used as bonded reinforcements for civil engineering structures [1–3]. This solution makes it possible to adapt many existing structures to more severe conditions (increased traffic or seismic risks). It is also used in the repair and rehabilitation of structures to extend their life-service. This solution is more economical than the construction of a new structure and limits traffic interruption. It also tends to decrease environmental impacts that such a repair project could cause. Yet, the mechanical performance of bonded joints can be compromised due to the presence of adhesion defects or damage such as voids, porosity, poor adhesion, or low cohesion strength [4–10]. Those defects are mostly internal to the joint structure and are not visible. It is therefore necessary to use a non-destructive technique to assess the quality of the adhesive joint.

Several works has been conducted to evaluate the adhesion levels of joined assemblies using different nondestructive techniques, such as IR thermography [11–13], optical methods (holography and shearography) [14], radiography (RX,Rγ) [15], or using ultrasonic testing methods including acoustic emission (EA) [16], bulk and guided (longitudinal, shear and Lamb) waves ultrasound (US) [17–24]. US methods show good potential for the characterization of the adhesive joint's properties. In their study, Korzeniowski et al. [23] evaluated the suitability of the pulse-echo method to detect voids of different shapes and sizes present in assemblies composed of a methacrylic resin-based structural adhesive,

between 60 mm by 60 mm steel sheets and being 1 mm thick. Castaings [19] used SH-guided ultrasonic waves to evaluate the mechanical properties of a bonded joint between two aluminum plates with different degrees of adhesion subjected to shear stresses. He materialized the different degrees of adhesion by neglecting the surface treatment on a test specimen and by covering an oily agent at the adhesive/adhesion interface on another test specimen. The study concludes that the technique offers good potential for assessing the level of interphase degradation. Li et al. [24] were able to detect a failure of adhesion by estimating ToF (time-of-flight) of the antisymmetric modes of Lamb waves. This type of defect was simulated in their steel/composite bonded test specimen with a Teflon insertion. They were able to correlate the decrease in ToF with the increase in the size of the defect. Attar et al. [18] studied the quality of bonding in a metal/adhesive/carbon-epoxy composite structure using Lamb guided waves. Yilmaz et al. [20] compared the performance of immersion, air-coupled, and contact ultrasonic testing techniques in bonding quality evaluation. Their tested samples were aluminum-epoxy-aluminum single lap adhesive joint containing debonding. These studies demonstrate a good potential of the US method to detect defects of void types, but encounter locks concerning other types of defects such as kissing-bonds. This conclusion is all the truer especially when the latter are materialized by an insertion of viscous matter. Additionally, the viscous matter is more representative of the actual surface contamination conditions that may induce a kissing-bond on site.

Yet, it is difficult to control FRP reinforcements adhesively bonded on concrete structures [25,26]. For such a structure, operators have indeed access to only one side for the measurements. In such conditions, the acousto-ultrasonic (AU) technique is one of the NDT methods that shows good potential to diagnose the bonded joint. This diagnostic aims at detecting and locating a defect, determining its size, classifying its type, and evaluating its severity. It combines an active phase (AU emission) and a passive phase with acoustic emission (AE) monitoring, getting the benefit of both AE and US techniques. The ultrasonic wave is emitted through the studied material and received after its propagation using piezoelectric transducers. The received signal is then analyzed using the AE method by processing the associated acoustic parameters. Thus, this technique works in a frequency domain similar to the one used in AE. Yet, to be able to detect damage, it is needed to dispose of the baseline data or the reference signal corresponding to the healthy state. Damage detection is carried out by comparing the parameters of the signals received from the tested samples with those from the reference signals.

The acousto-ultrasonic (AU) method was used by several authors [27–38] to detect and assess defects in assemblies. Wang et al. [30] evaluated matrix cracks in cross-ply Carbon Fiber Reinforced Polymer (CFRP) laminates using linear and nonlinear acousto-ultrasonic methods. Tanary et al. [30] used the acousto-ultrasonic technique to evaluate mechanical performance, such as the shear strength of composite single lap bonded joints. Kwon and Lee [34] investigated the correlation between the number of artificial defects and the acousto-ultrasonic parameters in CFRP-aluminum joints. This correlation could be established mainly with frequency parameters based on a spectral study of the detected signals. Barile et al. [35,36] used AU measurements to characterize the interlaminar strength of CFRP laminates in longitudinal and transverse directions, and to assess the detection of artificially induced impact barely visible on CFRP specimens. Zhang et al. [38] used the AU method combined with phase-shifted fiber Bragg grating to evaluate damage and delamination within complex composite structures. Vary and Lark [34] also used this technique to estimate the mechanical characteristics (tensile and shear strength) of a multi-layer composite (8-ply) with different fiber orientations on each layer.

The defect identification problem can be addressed using model-based approaches with a physical model, or by data-driven approaches based on a statistical model. The second approach is used in this work. The present work is dedicated to the application of three developed methodologies for the detection and classification of defects in adhesively bonded joints between CFRP laminate and RC beam using features extracted from AU signal. The main goal of this work is to present a robust methodology allowing the detection

and classification of the different bonded joint states. The approach should maintain or increase reliability and robustness in the treatment of data for analyzing the state of the bonded joint. In previous works [27–29], the authors used a methodology for defect classification on model specimens using mono-parametric analysis, PCA, and classification with a random forest approach. Their results showed a critical comparison between these methods for the defect detection and its identification. There are still challenges to overcome to reach a mature application level on site.

This paper deals with the application of the methodology developed in [28] onto large-scale specimens consisting in RC beams with adhesively bonded composite laminates. The objective of this study is to evaluate the robustness of the methodology regarding the combination of multiple defects in a single-lap adhesive joint, and the increase of substrate thickness. Combinations of three types of defects are investigated: voids, cohesive defects, and kissing-bond defects. The paper is structured as follows. The sample characteristics and experimental set-up are described in Section 2. In Section 3, we discuss the detection and the identification of all studied defects. The main conclusions are summarized in Section 4.

## 2. Materials and Experimental Set-Up

### 2.1. Samples

In this investigation, we studied reinforced concrete beams (C35/45 and 3400 mm × 200 mm × 200 mm dimensions) on which we bonded FLT S512 composite plates (3000 mm × 50 mm × 1.2 mm dimensions). This composite is unidirectional and constituted of T700 carbon fiber and epoxy matrix. It is pultruded with 60% of fiber content. Prior to their assembly, we increased the adhesion of both substrate surfaces by removing the brass on the concrete surface by sanding and performing a slight abrasion, followed by acetone cleaning of the composite plate. A two-component cold curing epoxy resin (Sikadur30), with an elastic modulus of 12.8 GPa, was employed as adhesive. For a good polymerization of the adhesive, a minimum of one week has been set before taking the measurements. The mechanical characteristics of the plates, adhesive joint, and beams, are summarized in Table 1.

**Table 1.** Composite, concrete beam, and epoxy properties.

| | Longitudinal Young's Modulus (GPa) | Transversal Young's Modulus (GPa) | Longitudinal Poisson's Ratio | Longitudinal Shear Modulus (GPa) | Transverse Poisson's Ratio | Density (g/cm$^3$) |
|---|---|---|---|---|---|---|
| Beam (C35/45) | 34.1 | 34.1 | 0.2 | - | - | 2.5 |
| Composite (FLT S512) | 160 | 12.3 | 0.25 | 5.02 | 0.25 | 1.6 |
| Adhesive joint- epoxy (Sikadur 30) | 12.8 | 12.8 | 0.29–0.34 | - | - | 1.95 at 20 °C |

We realized for a first campaign three samples (beams) with these materials (Figure 1). On each sample, we materialized two different defects located at 75 cm from the extremities of the composite plate. We also made sure on each beam to leave a healthy central area, in addition to the two areas with defects. As shown in Figure 2, each sample was divided onto three investigated zones: a zone without a defect ($\alpha$-type zone) at the center of the sample, and two zones with defects. In Table 2, we listed the different studied samples regarding their implemented defects and their average adhesive thickness (over 18 measurement points).

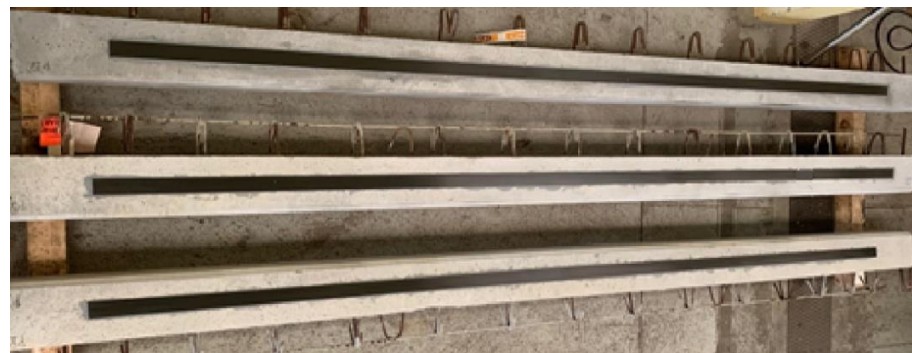

**Figure 1.** Samples of composite/RC-beam adhesively bonded assemblies.

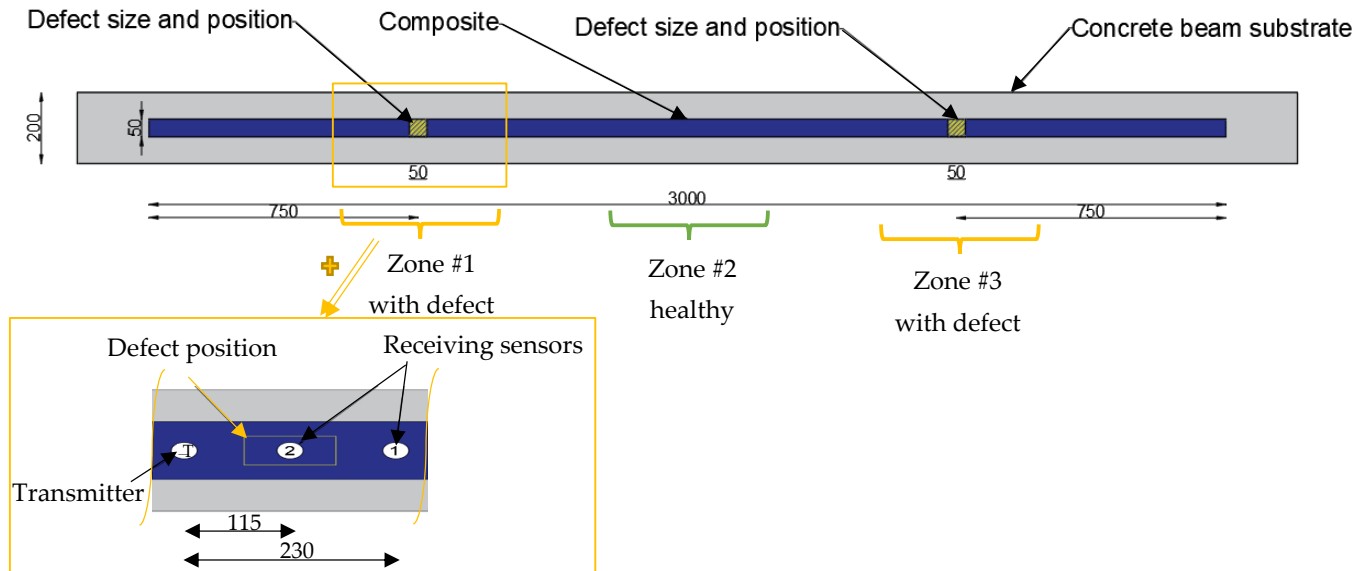

**Figure 2.** Schematic representation of the composite/beam assemblies with the location of the defects and sensors (T: transmitter, sensor #2 and sensor #1 denote the position of the receivers during the tests).

**Table 2.** Defects' sizes on different samples.

| | | | Voids: β-Type Zone | PU: γ-Type Zone | Kissing Bond: δ-Type Zone | Adhesive Average Thickness (mm) |
|---|---|---|---|---|---|---|
| 1st campaign | | Beam #1 | $50 \times 50$ mm$^2$ | $50 \times 50$ mm$^2$ | - | 0.94 |
| | | Beam #2 | - | $50 \times 50$ mm$^2$ | $50 \times 50$ mm$^2$ | 0.9 |
| | | Beam #3 | $50 \times 50$ mm$^2$ | - | $50 \times 50$ mm$^2$ | 1 |
| 2nd campaign | | Beam #4 | $50 \times 50$ mm$^2$ | $50 \times 50$ mm$^2$ | $50 \times 50$ mm$^2$ | 0.79 |
| | | Beam #5 | - | - | - | 0.86 |
| | | Beam #6 | $25 \times 25$ mm$^2$ | $25 \times 25$ mm$^2$ | $25 \times 25$ mm$^2$ | 0.84 |

To reproduce the most encountered defects, we investigated four types of zone:

- Reference zones with no defect in the adhesive joint (α-type zone). Guaranteed by strict compliance with the surface preparation conditions listed above.
- Zones with voids in the joint (β-type zone).
- Zones with the incorporation of polyurethane resin (8 MPa elastic modulus) on the whole thickness to materialize a poor cure defect or a softening of the resin (γ-type

zone), which could be, for example, due to ageing, with the presence of high moisture or poor cure.

- Zones with a lack of adhesion (kissing bond) between the epoxy joint and the composite substrate (δ-type zone). The adherent surfaces were partially contaminated with grease before applying the adhesive to create these weak interfaces.

### 2.2. Acousto-Ultrasonic Technique

The signals' emission is generated by an ARB-1410 card, with a high precision of 14-bit and a high speed of 100 MSample/second. The WaveGen software associated with this card was used to generate a one-cycle square signal with a frequency of 150 kHz and an amplitude of 10 Volts The emitting sensor is a piezoelectric S9204 with a bandwidth from 20 to 1000 kHz centered at 150 kHz. For the reception's system, we used a PCI-2 AE System card with 2-channel data acquisition coupled with two piezoelectric R15 sensors (bandwidth 50–200 kHz, resonance 150 kHz) and operated by AEWin software (Figure 3).

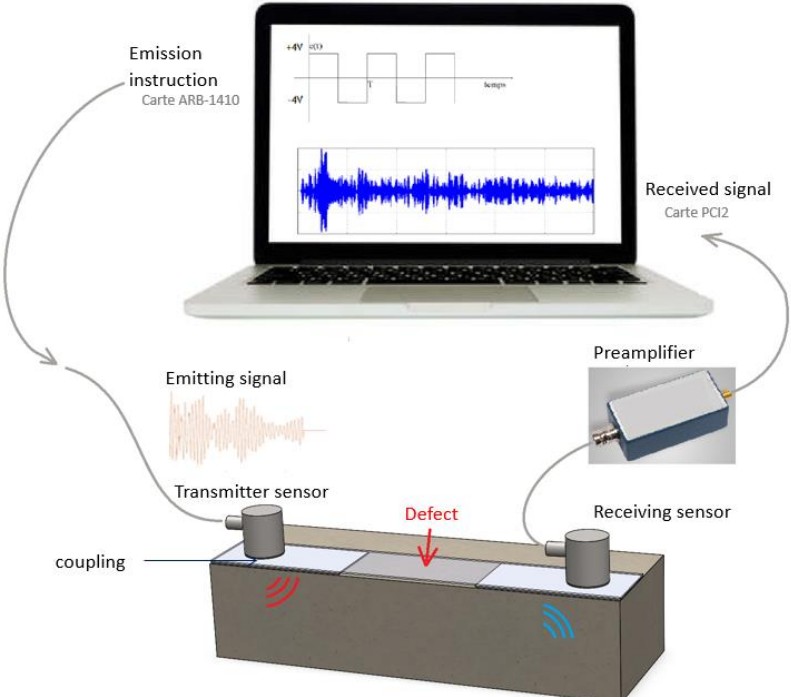

**Figure 3.** Experimental setup for acousto-ultrasonic measurements.

The sensors are coupled on the surface of the composite plate using a low-melting crystal, phenyl salicylate. They are placed aligned along the composite fiber direction: sensor #1 (receiver) is located at 230 mm away from the sensor T (transmitter), and sensor #2 (second receiver) is located at the center of the investigated zone. For the zones with defects, sensor #2 is located at the epicenter of the defect. We also use an IL40S-HT bandwidth preamplifier (32–1100 kHz) between the receiving sensors (#1 and #2) and the PCI-2 card, allowing us to acquire signals with a gain of 40 dB. We set the recording threshold at 45 dB.

For repeatable AU measurements, test conditions such as temperature, sensor location, spacing, and the type of coupling are kept constant. We conducted seven independent measurements on each zone to simulate the effects of the coupling procedure, and to measure the obtained dispersion. These measurements are carried out removing completely the sensors before repositioning them again. This implies a cleaning of the sensors and the specimen surface between each measurement. Their repeatability was assessed by a Pearson cross-correlation calculation between the signals received in the same studied zone. Those signals were recorded with a sampling frequency of 5 MSPS over a maximum length of 15 k points.

### 2.3. Defect Detection and Classification Methodology

In previous works [28], the authors used a methodology for defect detection and classification on a model specimen based on the combination of mono-parametric analysis, PCA, and classification with a random forest approach. Each signal is described by 10 descriptors or features: eight features from the time domain, and two from the frequency domain (amplitude, duration, rise time, counts, energy, counts to peak, peak frequency, centroid frequency, signal strength, absolute energy).

The damage detection and identification approaches in the mono-parametric analysis and the PCA are based on the separation of the collected data on two different sensor positions. The PCA has the objective to reduce the space's size by distorting reality as low as possible. In our case, we tried to reduce the 10 parameters in two or three dimensions by normalizing and centering the data.

An important advantage of using the random forest classification in our study is based on the use of the data collected from all the sensors. The random forest is a robust machine-learning algorithm consisted of two phases: a training phase, and a testing phase. During the training phase, a set of data is used to build the model consisting of multiple decision trees, a forest formed by T trees. From the training set, T bootstraps (resampled datasets with replacement) are then built. These are of the same size as the learning data but consist of randomly selected signals from the library. A tree consists of several nodes where the data are separated. At each node, n descriptors such as amplitude, frequency, . . . are randomly chosen from the set of descriptors. The algorithm will then determine which descriptor permits the optimal separation of the data. In our case, a model is built using the data recorded by sensors during the experiments on the first three beams (#1, #2 et #3). During the testing phase, another set of data is used to evaluate the classification error rate of the model. The testing then proceeds, with other measurements realized on the same beams. Thus, at each node of the trees, a criterion on the value of a descriptor is determined. Signals that are not assigned to a class at this node pass to the next node, where another n-descriptor is drawn, and so on until all signals have been separated. To label a dataset, each unlabeled signal traverses the T trees. Each tree issues a vote, the signal is assigned to the class that is most represented among the T votes. This algorithm has the benefit of being very fast. The testing is proceeded with other measurements realized on the same beams. Then, for more accuracy, we tested measurements realized on the other beams (#4, #5 et #6), which were not used for the training of the model.

## 3. Results and Discussions

### 3.1. Detection of the Defects

In Figures 4 and 5 we represent the signals received in the different zones of our assemblies using two of the most discriminating AE parameters. The comparison between healthy zones and zones with defects for the first three reinforced beams (Bm) was conducted via box and whiskers plots, in Figures 4 and 5 for sensor #1 and #2, respectively. The central mark on the box indicates the median, and the bottom and top edges of the box indicate the 25th and 75th percentiles, respectively.

We conducted the analysis of these figures considering the data from each beam separately. The data on one beam corresponds to three successive boxes, the first one concerning the signals received in the healthy zone and the other two, the signals received in the zones with defects. The figures show the dashed lines representing the detection criterion set by the methodology (20% deviation from the healthy zone) whose red color represents beam #1 (Bm1), while blue represents beam #2 (Bm2) and yellow represents beam #3 (Bm3).

We notice in Figure 4a that for Bm1 and Bm3, all the studied defects are detectable using the signal strength. For Bm2, combining cohesion ($\gamma$-type) and adhesion ($\delta$-type) defects, we have a maximum difference of $-10.5\%$ between the $\alpha$-type zone and the $\gamma$-type zone for this parameter. We note that to detect defects in this beam according to the

criterion we have set, the rise time can be used (Figure 4b). For this parameter, we observe a difference of 37.5% for the γ-type defect.

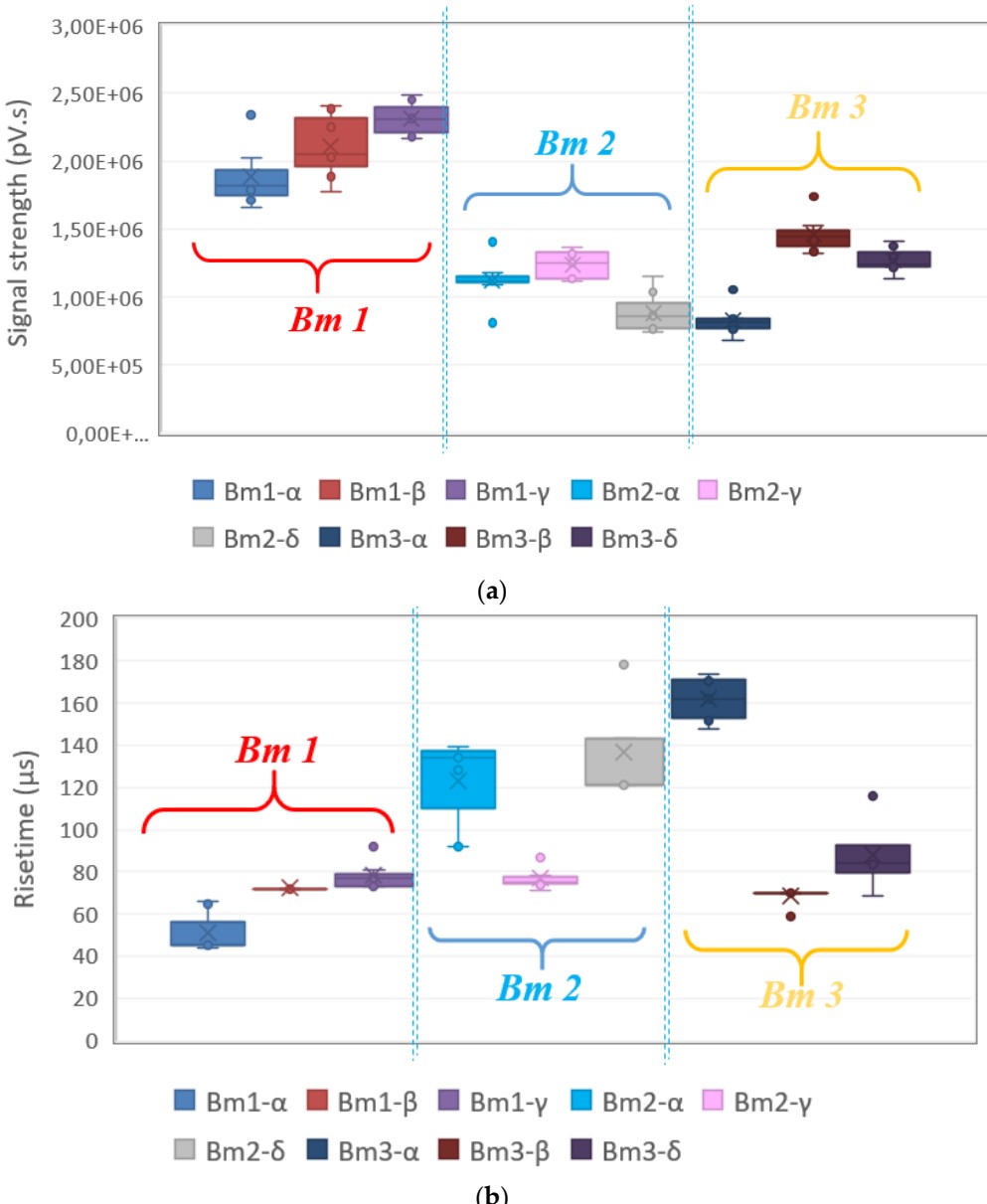

**Figure 4.** Parametric analysis of the signals recorded by sensor #1. (**a**) Signal strength (pV·s): Integral of the rectified voltage signal over the duration of the AE waveform and (**b**) risetime (μs): time between the first threshold crossing and the peak amplitude of the signal.

For sensor #2 located at the epicenter of defects (Figure 5), we note that on the first two beams (Bm1 and Bm2), the studied defects are detectable using the signal strength (Figure 5a) as parameter. This is not the case, however, for the void in the third beam, where we obtained a difference between α-type and β-type zones of 18% for this parameter. This defect in this beam is, however, well detected using the rise time for which we observe a difference of 31.2% between α-type and β-type zones. We notice, however, that the rise time is very dispersed for this sensor position in Bm3.

Using a simple mono-parametric criterion, we succeeded in detecting all the defect combinations studied for both sensor positions, as in [28]. It should be noted, however, that depending on the combination of defects, the suitability of the parameters to be used as indicators varies, and also depends on the position of the sensor. We also note that the

parameters considered as being the most relevant in [28], such as the peak frequency, are not necessarily relevant for this application. In practice, it is difficult to find one universal feature that is sensitive to all defects and for any geometry.

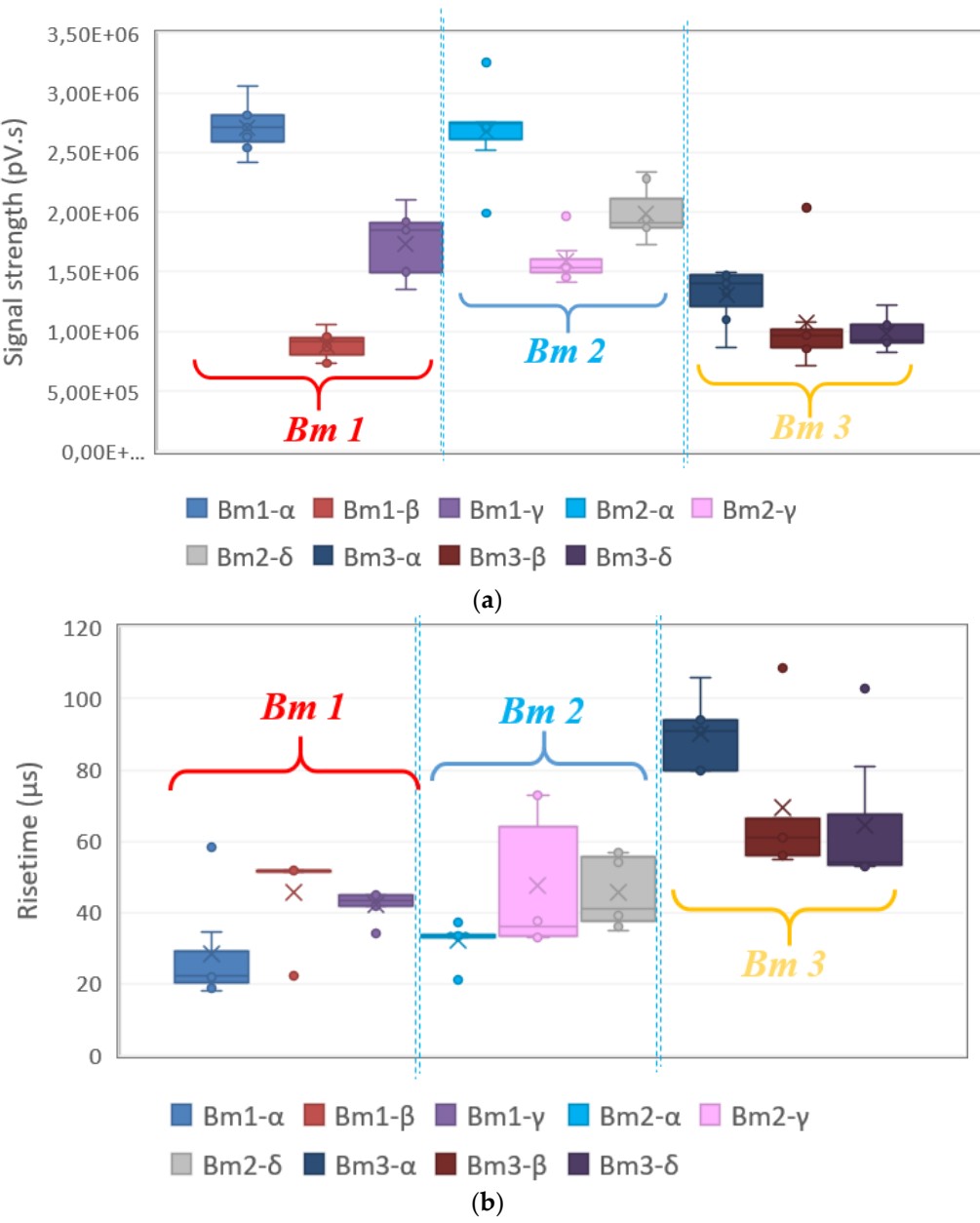

**Figure 5.** Parametric analysis of signals recorded by sensor #2. (**a**) Signal strength (pV·s) and (**b**) risetime (μs).

However, the boxes corresponding to the defect are not distinctive one from another. Thus, it seems difficult with this methodology to determine a simple criterion for the identification of the defects. More sophisticated methods are necessary for this purpose, such as PCA, and random forest.

### 3.2. Identification of the Defects

#### 3.2.1. PCA

The received signals can be analyzed using a Principal Component Analysis (PCA) applied to AE parameters. The PCA is conducted considering the data from each beam separately for the two sensors' positions. The signal distributions in the PCA plane for

the first three beams are respectively illustrated in Figures 6 and 7 for sensor positions #1 and #2. The data are described by the 10 parameters. For all PCA, we chose principal components to achieve a minimum 80% of the variance.

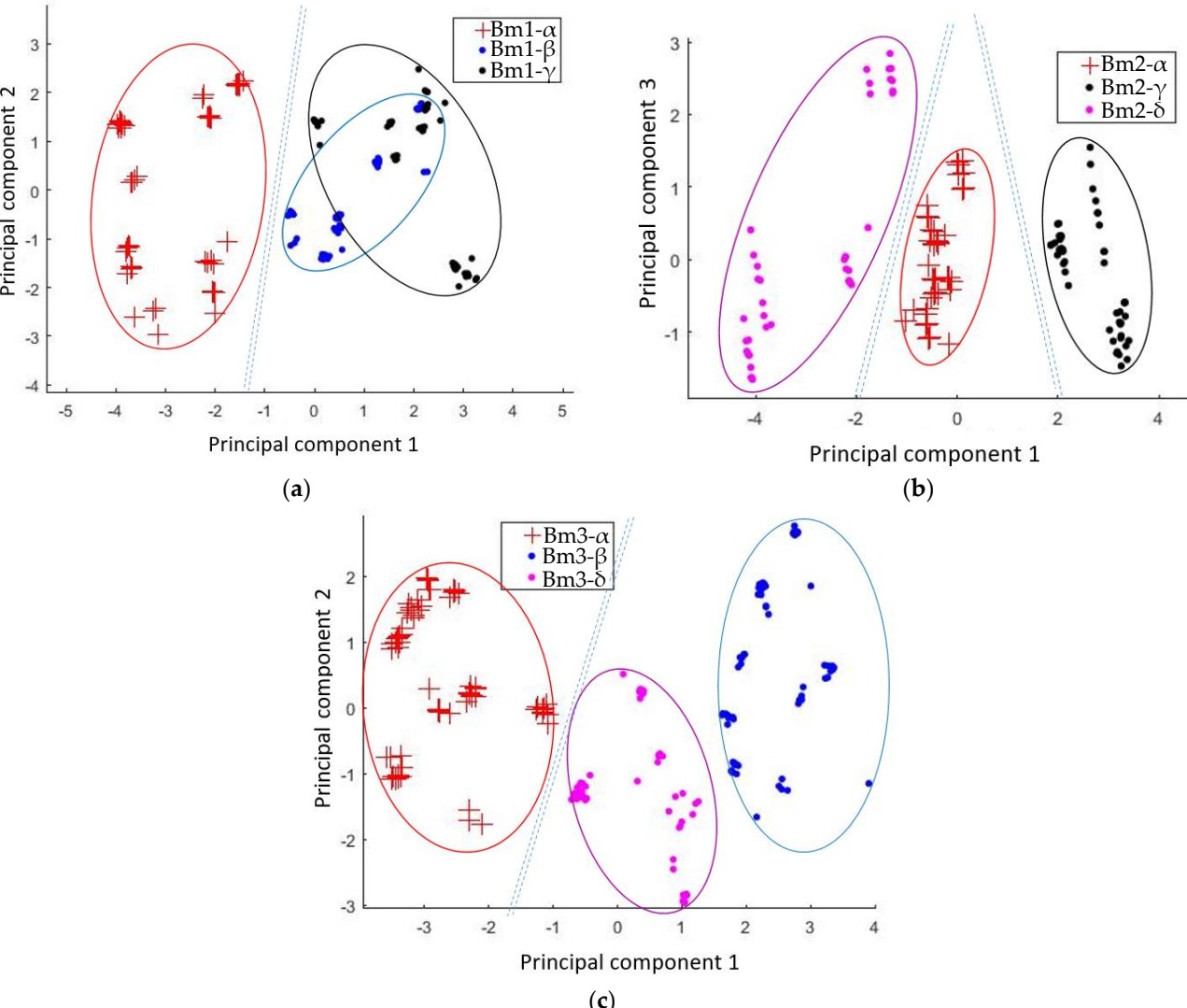

**Figure 6.** First principal component as a function of second principal component for signals received in sensor #1 (**a**) beam #1 (**b**) beam #2 (**c**) beam #3.

Figure 6 shows globally for each beam a separation of the clusters formed by the signals received on the healthy zones (red cross) from the clusters of the signals received on the zones with defects. Thus, we can conclude that the PCA makes it possible to detect the presence of a defect for sensor position #1 on the three beams.

On beams #2 and #3 (Figure 6b,c), we record a good separation of the clusters of signals received in zones with defect. This result is quite contrary in the case of beam #1 (Figure 6a). Thereby, for this sensor position, we could identify defects according to the beam where it is located.

In comparison with the data collected for sensor #1, we find for sensor #2 a better distinction between clusters for beam #1 (Figure 7a). For beam #2 (Figure 7b), the clusters of the signals collected from the zones with defects are also clearly discernable one from another, and from the cluster of signals collected on the healthy zone. The identification of the defects in these beams therefore seem clearly possible for this sensor position. We can note for beam #3 (Figure 7c) that the cluster formed by the signals received on the zones

with a defect of adhesion (dots in magenta) is not obviously discernable from the cluster of collected signals from the zone containing a void (blue dots).

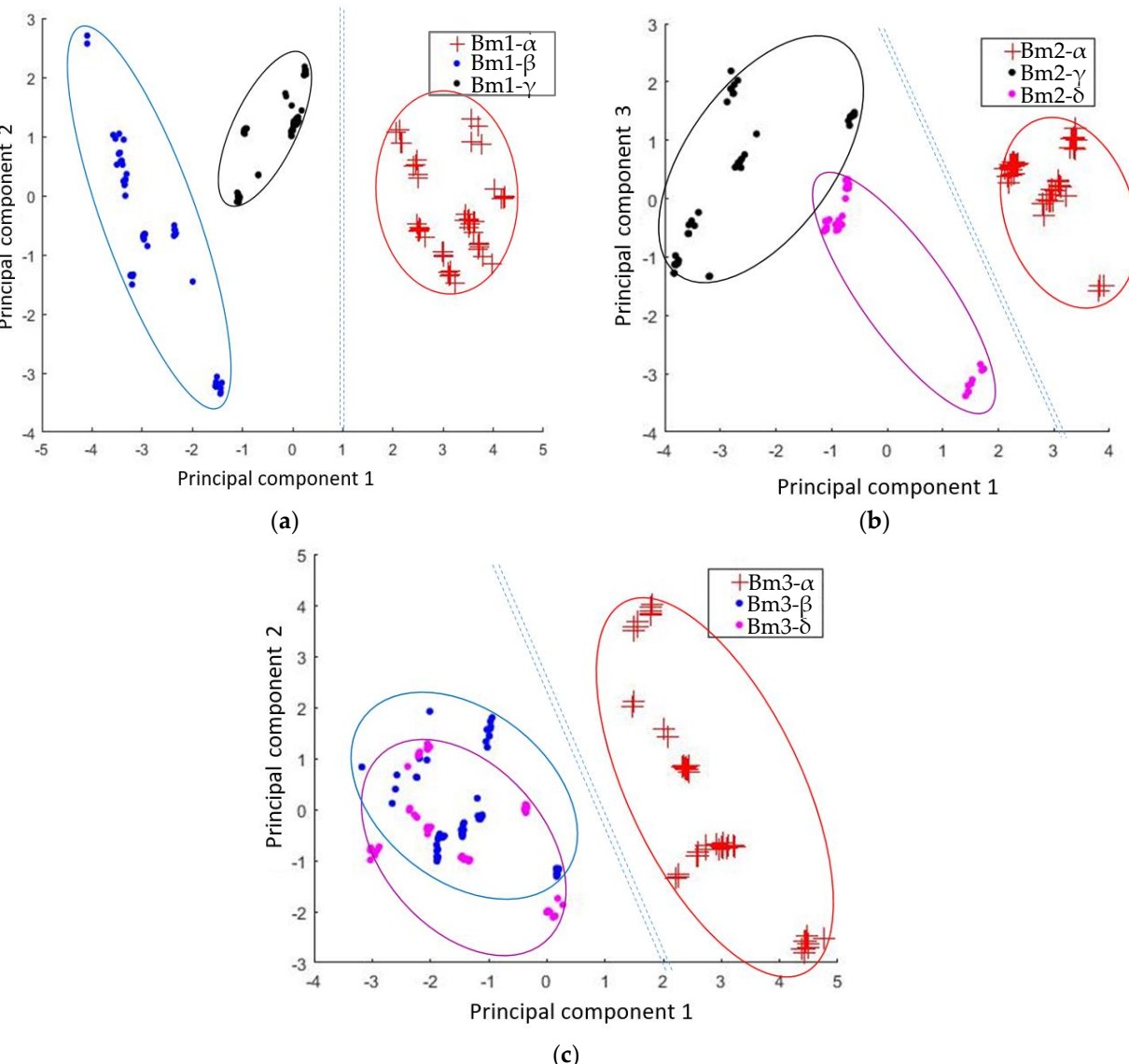

**Figure 7.** First principal component as a function of second principal component for signals received in sensor #2 (**a**) beam #1 (**b**) beam #2 (**c**) beam #3.

Though the scale is different, this result is similar with the study realized on small-scale assemblies with similar materials [28]. Indeed, in [28], for this sensor located at the epicenter of the defect, we found that the δ-type defect was more difficult to identify using PCA.

The first method of identification consisting in an analysis of the main components reveals a possibility of identifying, more or less clearly, all the simulated defects when we analyze results for each beam, and depending on the position of the sensor. Contrary to the conclusion of our previous article [28], it is impossible in this case study to fix a common reference for the three beams. This assertion is confirmed by the results of the overall PCA carried out for the nine zones of the first three beams. When analyzing the data obtained from all these zones simultaneously, it is indeed not straightforward to establish a clear separation between the zones without defect, and those containing defects (Figure 8). These results show the need to use a stronger method, such as random forest, for accurate classification of defects.

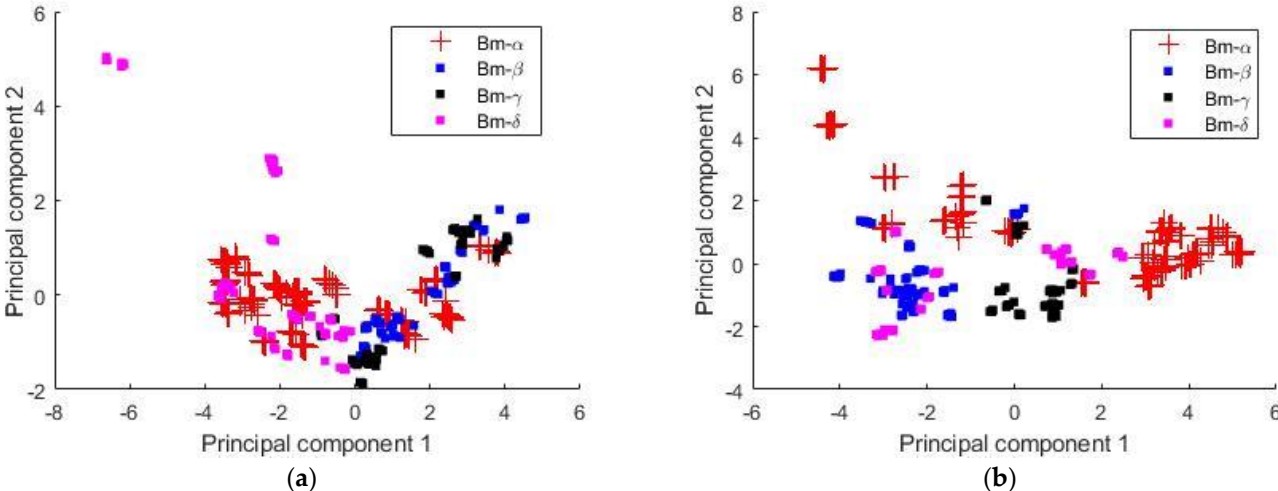

**Figure 8.** First principal component as a function of second principal component for signals received in all studied zones (**a**) sensor #1 (**b**) sensor #2.

### 3.2.2. Random Forest Classification

First Campaign of Classification

This analysis method of the received signals consists in setting a supervised classification using a decision tree forest (RFCAM, [39]). This approach first requires a training step based on the data acquired by both receiver sensors simultaneously. The classification method is then applied to signals different than those used for the training stage disregarding the sensors' position.

During the training stage, we constructed our classification model according to four classes (healthy, void, PU and kissing bond). For the "healthy" class, we used two series of measurements carried out on each healthy zone of the first three beams (1st campaign), and for each defect class, three series of measurements performed on each zone with defect. Therefore, each class is generated using six series of measurements (two series of measurements of the three healthy zones and three series of measurements of the two zones with the same type of defect). We generate a model consisting in 200 trees with a random selection of three descriptors on each tree's node. The remaining five series of measurements per healthy zone, and the four series of measurements per defective zone, are used to test the model on the first campaign. They constitute our testing data set.

Table 3 shows the results of the classification of the received signals over the different studied zones in the first campaign. In each row, the signals of the testing data set have been stored on the corresponding classes. We found that a large number of the studied signals are well labeled in their respective classes, which gives us overall recognition rates up to 95% (Table 4). The δ-type zone of beam #3 has the lowest recognition rate by majority vote (i.e., 49%). However, for this case, we note that the next highest class is the "PU class", which contains only 23% of the signals. In view of these results, it is then possible to accurately identify this defect without great difficulty. We note in this study that classification errors are most often in favor of a class whose defect is present in the same bundle. For example, we have the four signals received on the β zone of beam #3 that are incorrectly stored in the class "kissing bond" while for the same type of defect but which is on beam #1, we recorded the majority of classification errors on the "PU Class" with the 18 signals stored there. This observation seems to confirm that the propagation conditions in the beam has a major influence on the signal. This would imply that the selection of the reference zone (α-type) of our measurements must always be reset and could constitute a major bias for the extension of the technique on site.

**Table 3.** Classification results from majority voting rule of the signals of the testing stage on the 1st campaign.

|  |  | Class "Healthy" | Class "Void" | Class "PU" | Class "Kissing Bond" |
|---|---|---|---|---|---|
| Testing data set from beam #1 | Healthy α | **70** | 27 | 3 | - |
|  | Void β | - | **61** | 18 | 2 |
|  | PU γ | - | 20 | **64** | - |
| Testing data set from beam #2 | Healthy α | **89** | - | - | 15 |
|  | PU γ | - | 10 | **50** | 20 |
|  | Kissing bond δ | 17 | - | - | **65** |
| Testing data set from beam #3 | Healthy α | **52** | 15 | 10 | 24 |
|  | Void β | - | **78** | - | 4 |
|  | Kissing bond δ | 9 | 14 | 19 | **40** |

**Table 4.** Recognition and error rates corresponding to classification results.

|  |  | Classification Rate | Error Rate |
|---|---|---|---|
| **Testing data set from beam #1** | Healthy α | 70% | 30% |
|  | Void β | 75% | 25% |
|  | PU γ | 76% | 24% |
| **Testing data set from beam #2** | Healthy α | 86% | 14% |
|  | PU γ | 63% | 37% |
|  | Kissing bond δ | 79% | 21% |
| **Testing data set from beam #3** | Healthy α | 51% | 49% |
|  | Void β | 95% | 5% |
|  | Kissing bond δ | 49% | 51% |

Second Campaign of Classification

During the second campaign of classification, we tested the model with the data recorded on new specimens that did not particpate in the training of the model (beam #4, beam #5 and beam #6). Table 5 shows the results of the classification of the received signals over the different studied zones on these three beams of the second campaign. In this case, the identification of the defects proved to be more difficult. However, we detetected all the studied defects of the same size (those of the beam #4) as the ones used to train the model. The γ-type defect's identification is shown to be the most occurate in our study with a classification rate of 47%.

**Table 5.** Classification results from majority voting rule of the signals of the testing stage on the 2nd campaign.

|  |  | Class "Healthy" | Class "Void" | Class "PU" | Class "Kissing Bond" |
|---|---|---|---|---|---|
| Testing data set from beam #4 | Void β | - | **78** | 84 | 7 |
|  | PU γ | 62 | 11 | **75** | 11 |
|  | Kissing bond δ | 56 | - | 98 | - |
| Testing data set from beam #5 | Healthy α | **72** | - | - | 72 |
| Testing data set from beam #6 | $\frac{1}{4}$ Void β | 3 | **48** | - | 98 |
|  | $\frac{1}{4}$ PU γ | 10 | 75 | **46** | 11 |
|  | $\frac{1}{4}$ Kissing bond δ | 116 | 1 | 18 | **6** |

On beam #5 free of defects, the healthy zone is barely recognized. This result confirms that it is crucial to assign a new reference area each time for any new study. For smaller defect sizes (beam #6), they cannot be recognized correctly. The β and γ type defects are still detected as a defect, which was not the case for the δ type defect.

## 4. Conclusions

The main purpose of this article was to evaluate the robustness of the diagnosis methodology (detection/identification for three types of joint defects) implemented in a previous study [28], when applied to large-scale specimens (concrete beams reinforced by adhesively bonded composite plate).

The mono-parametric analysis applied to these measurement data revealed that for all the performed defect combinations, the method remains effective in detecting these defects per beam. However, the parameters relevant for this detection vary one beam from the other and the defect influence is not always reproducible. Thus, for the position of the sensor located at 230 mm from the transmitter (sensor #1), the energy parameter shows rather upward values in the presence of defect (contrary in [28]).

With a PCA method applied per beam, defect identification is possible depending on the position of the sensor. We have seen that, depending on the studied beam, we can have a good distinction of clusters for one sensor position and not for the other. Thus, for beam #1 having the couple of defects void-cohesion, the defect's clusters are not discernible using sensor #1. For beam #3, combining the void and kissing bond defects, the clusters are also not discernible for sensor #2 located at the epicenter of the defect.

Using the random forest classification method, we are able to identify all defects combining the data from the two sensors for all the beams. Nevertheless, we obtained lower recognition rates than those found in [28]. Moreover, we noted that the recognition rate of a type of defect was variable depending on the size of defects.

In general, we note a slight decrease in the effectiveness of this methodology in detecting and identifying the studied defects due to the larger scale specimens (beams), the presence of multiple types of defects on the same beam, and the influence of the propagation conditions of ultrasonic waves. However, compared to commonly used techniques such as IR thermography, the AU technique show better potential to evaluate the bonded joints quality. Nevertheless, a global analysis is not yet possible due to the fact that each type of defect must be analyzed using its own 35 reference beam, a common reference having not been able to be defined because of the different propagation conditions between the beams. This could also affect the enrichment of the learning library for supervised classification. It is therefore important to quantify the influence of the propagation conditions on the AU signals with the help of a modelling work in order to improve the proposed methodology. Additionally, it should be interesting to evaluate the acousto-ultrasonic activities under mechanical testing on the assemblies.

**Author Contributions:** Conceptualization, C.A.T.S., S.C., L.G. and N.G.; methodology, C.A.T.S., S.C., L.G. and N.G.; software, C.A.T.S.; validation, C.A.T.S., S.C., L.G. and N.G.; formal analysis, C.A.T.S.; investigation, C.A.T.S.; data curation, C.A.T.S., writing—original draft preparation, C.A.T.S.; writing—review and editing C.A.T.S., S.C., L.G. and N.G.; supervision, S.C., L.G. and N.G.; project administration, C.A.T.S., S.C., L.G. and N.G.; funding acquisition, N.G., S.C. and L.G. All authors have read and agreed to the published version of the manuscript.

**Funding:** This work was supported by the Pays de la Loire region through a co-financing of a doctoral fellowship.

**Conflicts of Interest:** The authors declare no conflict of interest.

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
