# Peer review of "Defects Detection and Identification in Adhesively Bonded Joints between CFRP Laminate and Reinforced Concrete Beam Using Acousto-Ultrasonic Technique"

_jcs, doi:10.3390/jcs6110334_

Round 1

Reviewer 1 Report

The development of non-destructive testing methods is extremely important for assessing the bearing capacity of composite structures such as reinforced concrete strengthened with  CFRP laminate, since it allows avoiding mechanical impact and integrity violations. In this regard, the research topic deserves attention and corresponds to the direction of the journal. However, the effectiveness of the proposed method, as noted by the authors, has not yet been fully investigated. They "note a slight decrease in the effectiveness of this methodology in detecting and identifying the studied defects due to the larger scale specimens (beams), the presence of multiple types of defects on the same beam and the influence of the propagation conditions of ultrasonic waves. Nevertheless, a global analysis is not yet possible due to the fact that each type of defect must be analyzed using its own reference beam, a common reference having not been able to be defined because of the different propagation conditions between the beams". The disadvantage of the approach is that for evaluation it is necessary to have a reference beam. This greatly reduces the effectiveness of the method and its practical application. Also, authors don't consider the influence of the loads applied to the beam on the current value of the adhesion parameters, when evaluating the method robustness. However, on the other hand, the data presented in the article are of considerable interest and can serve as the formation of a data bank for a more effective analysis of the state of structures in the future. In this regard, I think that these should be published. There are several comments on the article that I would like to see taken into account in the final version of the article:

1. What do the data in tables 3 and 5 mean? Please give more detailed explanations.

2. Have the adhesion parameters been evaluated by any method other than ultrasonic? If it is not, it would be interesting to see a comparison of these data with the results of mechanical tests under load in future studies by the authors.

Reviewer 2 Report

In this work, the authors conducted defect detection and identification in adhesively bonded joints between CFRP laminate and reinforced concrete beam using acousto-ultrasonic technique, where the data-driven approaches were employed for defect identification. This work can be considered for publication after careful revision following the comments listed below:

(1) The fabrication procedure for the samples together with a schematic are suggested to be added in section 2.1.

(2) A picture or schematic for the experimental setup should be added with the introduction of the instruments used in the experiments.

(3) More detailed information for the defect detection and classification methods should be added in section 3.2. Although the authors claimed that the have conducted similar studies in ref. [28], it is better to present enough information in this work to let readers fully understand the methods used here without referring to the published work of the authors.

(4) It seems that the defect identification results are not satisfactory enough via principal component analysis and random forest classification. Is it possible to further improve the identification methods? Besides, it is suggested to compare with the identification results via other commonly used methods to demonstrate the advantages of the methods used in this work.

(5) Data-driven modeling was mentioned in abstract and introduction. However, in the following studies, it was not mentioned any more. Although data-driven modeling is a hot topic, the methods used here should not be defined as data driven modeling according to the reviewer. Please consider to make proper revision.

(6) For the measurement, did the author carry out repeated tests for one samples to make sure the reliability of the test data? For composites, it is better to do that.

Round 2

Reviewer 1 Report

Recommendations were taken into account.

Reviewer 2 Report

The authors revised their manuscript following me comments. This revised manuscript can be accepted for publication. Thanks.